# A Combination of Mediterranean and Low-FODMAP Diets for Managing IBS Symptoms? Ask Your Gut!

**DOI:** 10.3390/microorganisms10040751

**Published:** 2022-03-30

**Authors:** Arezina Kasti, Konstantinos Petsis, Sophia Lambrinou, Konstantinos Katsas, Maroulla Nikolaki, Ioannis S. Papanikolaou, Erifili Hatziagelaki, Konstantinos Triantafyllou

**Affiliations:** 1Department of Nutrition and Dietetics, ATTIKON University General Hospital, 12462 Athens, Greece; kastiare@med.uoa.gr (A.K.); kostas.petsis@hotmail.com (K.P.); sophialambrinou@gmail.com (S.L.); kostasktss1@gmail.com (K.K.); maroullanikolaki@gmail.com (M.N.); 2Institute of Preventive Medicine Environmental and Occupational Health Prolepsis, 15125 Athens, Greece; 3Hepatogastroenterology Unit, 2nd Department of Internal Propaedeutic Medicine, Medical School, National and Kapodistrian University of Athens, Attikon University General Hospital, 12462 Athens, Greece; ispapn@hotmail.com; 42nd Department of Internal Propaedeutic Medicine, Medical School, National and Kapodistrian University of Athens, Attikon University General Hospital, 12462 Athens, Greece; erihat@med.uoa.gr

**Keywords:** irritable bowel syndrome, low-FODMAP diet, inflammatory bowel disease, Mediterranean diet, inflammation, gut microbiota

## Abstract

Among other factors, food intolerance is cardinal in triggering irritable bowel syndrome (IBS) symptoms in a significant percentage of patients. As a result, specific dietary patterns are the first-line therapeutic approach. The low-FODMAP diet (LFD) is gaining ground as the most well-documented diet intervention that significantly reduces IBS symptoms. Though the LFD improves symptoms, the diet’s impact on intestinal low-grade inflammation, one of the cardinal mechanisms contributing to symptom development, remains doubtful. On the other hand, the Mediterranean diet (MedDiet) is recommended for chronic low-grade inflammation-related diseases because of its anti-inflammatory properties, derived predominantly from olive oil and phenolic compounds. Thus far, the role of a modified LFD, enriched with the MedDiet’s anti-inflammatory components, has not been evaluated in IBS patients. This review aims to examine the hypothesis of a potential combination of the immunomodulatory effects of the MedDiet with the LFD to improve IBS symptoms.

## 1. Introduction

Irritable bowel syndrome (IBS) is a highly prevalent chronic functional gastrointestinal disorder. The cardinal symptoms include abdominal pain, flatulence, bloating, and changes in bowel habits related to stool frequency and consistency, in the absence of detectable structural and biochemical abnormalities [1,2].

IBS is divided into four subtypes, depending on the predominant bowel habit alteration, according to the Rome IV criteria: IBS with constipation (IBS-C), IBS with diarrhea (IBS-D), IBS with defecation alterations in between constipation and diarrhea (IBS-M), and unclassified IBS (IBS-U). IBS symptoms may appear after an infectious disease, e.g., enteritis from viral (Norwalk), parasitic (*Cryptosporidium* spp. or *Giardia* (*Giardia duodenalis* or *Giardia lamblia*)), or bacterial (*Campylobacter jejuni* and *Salmonella*) infection (postinfectious IBS) [3,4,5]. The underlying mechanisms of postinfectious IBS (PI-IBS) pathogenesis, though not clearly understood, are potentially caused by residual inflammation or by alterations in mucosal immunocytes (B and T cells), mast cells, enterochromaffin-like cells, and the intestinal flora [4,6], as well as altered intestinal permeability (decreased expression of the tight junction-associated proteins Zonula occludens-1 (ZO-1) and α-cathenin) [7].

Gut commensal bacteria number a few trillion, with the predominant bacterial phyla *Firmicutes* (e.g., *Lactobacillus*), *Actinobacteria* (e.g., *Bifidobacteria*), *Bacteroidetes* (e.g., *B. fragilis*), and *Proteobacteria* (e.g., *E. coli, Yersinia, Salmonella*) living in a balanced relationship with viruses, fungi, archaea, protozoa, and the host. The beneficial bacteria *Lactobacilli* and *Bifidobacteria* have anti-inflammatory properties, protecting the barrier function, among other things [1]. In IBS, there is an imbalance between “good” and “bad” flora called dysbiosis, which has been associated with disruption of the mucosal enteric barrier, loss of immunotolerance, and low-grade inflammation limited to the intestine [6,8,9,10].

Low-grade mucosal inflammation represented by chronic inflammatory cells, such as mucosal enteroendocrine cells and T lymphocytes, that have been found in rectal mucosal biopsy specimens and visceral hypersensitivity are present in all IBS subtypes (including PI-IBS), providing evidence to the contemporary theory that IBS is not a functional disorder [11,12,13].

Up to 89% of IBS patients associate their intestinal distention and pain with food consumption. Additionally, almost 90% of them choose to eat less food to lessen postprandial discomfort; therefore, specific dietary patterns are used as a first-line therapeutic approach, showing beneficial effects [1,3]. Studies have shown that poorly absorbed dietary components are fermented by the intestinal bacteria, causing osmotic effects in the colon with gas production, pain, and diarrhea [14,15]. The acronym FODMAPs (fermentable oligosaccharides, disaccharides, monosaccharides, and polyols) is used by Monash University researchers to describe a family of poorly absorbed, osmotically active short-chain carbohydrates (SCCs) rapidly fermented by the gut bacteria. A low-FODMAP (LFD) diet significantly reduces IBS symptoms compared to a habitual Australian diet [16]; however, although the LFD improves IBS symptoms, its efficacy regarding the reduction in inflammation is not proven [17]. On the contrary, the well-known anti-inflammatory benefits of the Mediterranean diet (MedDiet) may occur due to olive oil as the principal source of fat, which contains at least 30 phenolic compounds [17,18]. Polyphenols have antioxidant, anti-inflammatory, and antimicrobial effects [17]. Consequently, the MedDiet is an effective dietary model for chronic low-grade inflammation-related diseases such as cardiovascular disease, atherosclerosis, metabolic syndrome, and obesity [19,20,21].

Considering the value of the LFD in improving symptoms in IBS patients, and the proven anti-inflammatory properties of the Mediterranean diet in many clinical conditions, we aimed to assess the immunomodulatory effect of the MedDiet combined with the LFD in managing IBS (Figure 1).

## 2. Materials and Methods

We performed a literature search in the PubMed and Cochrane databases for articles written in the English language. We used evidence from original articles written in the English language, excluding reviews, abstracts, conference presentations, pediatric studies, editorials, and study protocols. Given the well-known efficacy of the low-FODMAP diet in alleviating IBS symptoms, we used the terms “Mediterranean Diet”, “irritable bowel disease”, and “inflammation”, in order to discuss a potential extrapolation of the results of this literature search on the treatment of IBS with the combination of the two diets.

## 3. Results

The literature searches revealed no results. Thus, in the absence of data exploring the immunomodulatory effects of the MedDiet on IBS symptoms, we review the immunomodulatory effects of the MedDiet below, as well as the pathophysiologic correctness of combining the two diets for the management of this functional disorder.

### 3.1. The Effect of LFD on IBS Subtypes, Gut Microbiota, and the Immune System

The LFD focuses on limiting the intake of carbohydrates found in wheat-based products, onion, garlic, legumes, dairy, and many fruits and vegetables. These carbohydrates are incompletely absorbed in the small intestine and cause gas and water retention. Fruits and vegetables high in FODMAPs are often reduced but not replaced with low-FODMAP alternatives [22]. As a result, long-term use of the LFD, which decreases prebiotic intake, will probably have unpleasant effects on the gut microbiota [23]. Moreover, constipation is associated with low fiber intake, and the LFD further decreases fiber consumption [24]. Hence, treatment for IBS-C is targeted towards increasing fiber consumption and fiber supplements, in order to alleviate constipation, as recommended (Level II Evidence, Grade B) [1,25]. While the LFD relieves IBS-C patients’ symptoms to a lesser extent compared to other subtypes, it has been postulated that the LFD is a beneficial treatment regardless of the IBS classification. Marsh et al. indicated that the LFD clinically and significantly improved all symptoms, with IBS-C patients benefiting less [24]. Additionally, lower fiber intake correlated with decreased symptoms among IBS-D subjects, but this did not apply to the other IBS subtypes. In a recent meta-analysis, van Lanen et al. showed the efficacy of the LFD among all IBS subtypes. Patients with constipation as the predominant symptom and with a mixed stool pattern were similarly relieved and the symptoms subsided, but the results were not consistent among the studies. Furthermore, they underlined that the subgroups were small, and they concluded that more studies are needed to determine if the efficacy of the LFD is consistent among IBS subtypes [26].

Venter et al. highlighted the bidirectional relationship between diet and the immune system underlying the protective role of specific nutrients. Nutritional status may also have an effect on the gut microbiota and the intestinal mucous membrane [27]. Many reports have linked dysbiosis with IBS, referring to a significant reduction in the genera *Lactobacillus* and *Bifidobacterium* compared to healthy people [6,8,9,10]. The LFD reduces dietary prebiotics (fructans and GOS) that stimulate immunomodulatory bacteria (Bifidobacteria and Faecalibacterium prausnitzii) and interact with PRRs, protecting the intestinal barrier and epithelial cells [28] (Table 1). This effect is a limitation of the LFD given the barrier dysfunction in IBS-D and PI-IBS [29].

The restrictive phase of the LFD (4 weeks) led to dysbiosis in 42% of IBS patients, while 46% of them did not experience any dysbiosis [8]. A systematic review showed that increased intestinal permeability was present in 37–62% of IBS-D and 16–50% of PI-IBS patients and correlated with barrier dysfunction and diarrhea severity [31]. Intestinal permeability results in visceral hypersensitivity and reduces ZO-1 expression [32]. Endotoxins such as LPS are components of the outer membrane of Gram-negative bacteria which enter the lamina propria, inducing inflammation and increased cytokine expression. Inflammation leads to further elevation in blood endotoxins and circulating cytokines due to intestinal permeability as a result of mucosal damage [33,34] (Figure 2).

Until now, we believed that the LFD reduces the beneficial bacteria only in IBS patients with altered gut microbiota and dysbiosis. The same findings were reported in healthy volunteers, showing that the LFD decreases bacterial amounts overall and more specifically the abundance of Actinobacteria—mainly Bifidobacteria [30]. Investigators [35] suggest probiotic administration in parallel with the LFD to restore the loss of beneficial bacteria. The role of probiotics in bowel function and symptom relief in IBS patients is supported by more than 80 trials and approximately 10,000 patients [33].

Restricting fermentable substrates for saccharolytic gut bacteria reduces SCFA production. While SCFAs inhibit intestinal colonization by pathogenic bacteria, their reduction leads to a lower amount of substrate for colonocytes and an increase in microbes [25,29]. Long-term LFD consumption may induce adverse effects beyond colonocyte metabolism and micronutrient deficiencies [15,25]. Although there is not enough evidence, we may speculate that long-term LFD consumption leads to deficiencies similar to those from exclusion diets. Consumption of a gluten-free diet (GFD), for instance, excludes cereals (wheat, barley, rye) like the LFD and exposes patients at high risk for iron, folate, and zinc inadequacies. Fiber consumption in the LFD is restricted (including legumes and some fruits and vegetables). Thus, imbalances in nutritional status and symptom exacerbation are expected, particularly in IBS-C. Recently, a double-blind crossover trial was conducted in 26 patients with IBS, which were randomly assigned to one of three low-FODMAP diets differing only in the total fiber content (control 23 g/d; minimally fermented sugarcane bagasse 33 g/d; and the combination of sugarcane bagasse with resistant starch ∼45 g/d). The results showed that supplementation of fibers during the initiation of a low-FODMAP diet did not alter the symptomatic response in patients with IBS but augmented stool bulk and normalized low stool water content and slow transit, indicating the beneficial role of the combination of dietary fiber and a low-FODMAP diet in IBS symptom management [36]. Moreover, restriction of lactose-containing dairy products limits calcium intake and depletes vitamin D stores. Finally, the LFD contains low amounts of phenolic compounds, anthocyanins, and antioxidants—flavonoids, carotenoids, and vitamin C—which are components of fruits and vegetables (e.g., onions, garlic, cherries, blackberries) [37].

### 3.2. The Effect of MedDiet on the Immune System

The MedDiet’s characteristics include olive oil consumption, fruits, vegetables, cereals (mainly whole grain), legumes, and nuts. Fish, white meat, eggs, and fermented dairy products (cheese and yogurt) are also included but in moderate amounts. Small amounts of red meat and foods rich in sugars are permitted. Wine—preferably red—with meals is recommended in moderate quantity and frequency. Fat intake comprises 40–50% of daily calories with emphasis on monounsaturated fatty acids (MUFAs), mainly from olive oil (15–25% of calories), in contrast to saturated fatty acids (SFAs), which comprises less than 8% of the consumed daily fat. There is high intake of *omega*-3 fatty acids from fish and other seafood and plant sources, and the ratio of *omega*-6/*omega*-3 is low (2:1–1:1 compared to 14:1 in other European countries). In addition, high consumption of dietary fiber and antioxidant compounds may act together to produce favorable anti-inflammatory effects [19]. The MedDiet is characterized by an abundance of antioxidant/anti-inflammatory components, although their detailed descriptions are beyond the scope of this review. The predominant anti-inflammatory ingredients include *omega*-3 fatty acids, olive oil and phenolic compounds, and fibers.

#### 3.2.1. Omega-3 Fatty Acids

The polyunsaturated fatty acids (PUFAs) are *omega*-3 (α-linolenic acid (α-LA), eicosapentaenoic acid (EPA), docosahexaenoic acid (DHA)) and *omega*-6 PUFAs (linoleic acid (LA), arachidonic acid (AA)) [38]. Western-type diets are rich in *omega*-6 PUFAs in comparison with *omega*-3 PUFAs, with the consumption ratio reaching 20–30:1. *Omega*-3 PUFAs and their bioactive metabolites compete with *omega*-6 PUFAs to promote the resolution of inflammation [39,40]. EPA and DHA inhibit the activation of the transcription factor NF-κB (Nuclear Factor–B) in macrophages. They can impede leukocyte chemotaxis, adhesion molecule expression, and leukocyte–endothelial adhesive interactions. Eventually, they prevent eicosanoid production, inflammatory cytokine secretion (Tumor Necrosis Factor-alpha (TNF-a) and interleukin (IL)-6), T-cell reactivity, and phagocytosis [41,42].

#### 3.2.2. Olive Oil and Phenolic Compounds

Virgin olive oil (VOO) daily intake, the primary source of fat in the MedDiet, ranges between 25 mL and 50 mL. It is high in MUFAs, with the dominant lipid, oleic acid, found in proportions up to 83% [43]. The Western diet has been associated with high CRP, IL-6, vascular cell adhesion molecule-1 (VCAM-1), and Intercellular Adhesion Molecule-1 (ICAM-1) levels. Endothelial cells are activated in response to the inflammatory process to produce cytokines and adhesion molecules. Many studies have shown that VOO decreases plasma concentrations of IL-6 and CRP and suppresses the expression of VCAM-1 and ICAM-1. The ATTICA study showed that adherence to the MedDiet was associated with a reduction of 20% in CRP and 17% in IL-6 levels. Similar results were reached by the Nurses’ Health Study, indicating that high adherence to the MedDiet was associated with lower concentrations of inflammatory biomarkers (CRP, IL6) [19]. The VOO benefits are related to the components in the unsaponifiable fraction (about 2%), including phenolic compounds (absent in oils derived from seeds or fruits), phytosterols, and tocopherols. Phenolic concentrations in extra-VOO (EVOO) consist of various components (phenolic acids, lignans, flavones, flavone glycosides, phenolic alcohols, and secoiridoids) [44]. Over 90% of phenolic compounds constitute secoiridoids (oleuropein, oleocanthal, and ligstroside aglycone). Their alcohol derivatives, hydroxytyrosol (HT) and tyrosol, and flavonoids and lignans are considered potential agents against inflammation and exhibit inhibitory action on NF-κB [45]. EVOO interferes with arachidonic acid that represents the beginning of the inflammatory response. The first step is prostaglandin production mediated by cyclooxygenase (COX). Simultaneously, inflammatory cytokines induce NF-κB expression and NLRP3 inflammasome assembly. EVOO decreases COX concentrations and inflammatory cytokines and impairs the NF-κB pathway [43,46]. Molecular mechanisms of polyphenol anti-inflammatory properties further include inhibition of COX-2, Lipoxygenase (LOX), inducible Nitric oxide synthase (iNOS), and the activating protein-1 (AP-1), and activation of phase II antioxidant enzymes, mitogen-activated protein kinase (MAPK), protein kinase-C (PKC), and nuclear factor erythroid 2-related factor (NRF2) [47]. Furthermore, HT inhibits the activation of the TLR-4 and NK-kB pathways, which are associated with intestinal permeability [44]. Aromatic plants, spices, seeds, and nuts rich in polyphenols are abundant in the MedDiet. Polyphenols are divided into two subgroups, flavonoids and non-flavonoids [44]. Flavonoids have a protective role against cancer, gastrointestinal disorders, and other diseases [48]. In the group of phenolic compounds, there are, beyond HT, two main components: resveratrol (RSV) and quercetin (QUE) [44,49]. QUE inhibits the activation of the NF-κB pathway induced by IL-1β. In addition, it impedes TNF-induced IFN-γ-inducible protein ten and macrophage inflammatory protein two gene expression in the murine small intestinal epithelial cell line Mode-K [48]. RSV is a natural antioxidant with anti-inflammatory potency. RSV and its metabolites and conjugated products inhibit NF-κB and eicosanoid production, may block TLR4, proinflammatory genes, and IL-17, and activate NRF2. The effects of RSV on the gut microbiota have been studied in mouse models. RSV supplements decreased the *Firmicutes/Bacteroidetes* ratio, avoiding *Enterococcus faecalis* growth and achieving *Lactobacillus* and *Bifidobacterium* proliferation [44].

#### 3.2.3. Fiber

Fiber and carbohydrates are the energy source for gut bacteria, and their fermentation produces short-chain fatty acids (SCFAs). SCFAs lead to an increase in *Bifidobacteria*, *Bacteroidetes,* and *Akkermansia muciniphila*, with positive effects on barrier integrity, motility, and alleviation of inflammation [40]. Fiber intake in the MedDiet is two times higher than in a usual Western diet [40]. Fiber is divided into soluble and insoluble forms. Insoluble fiber captures water in the stool and increases gut peristalsis and stool bulk. It reduces the time of colonic fermentation, contributes to microbial diversity, and prevents inflammation. Soluble fibers are prebiotics with anti-inflammatory potential (fructooligosaccharides (FOS), galactooligosaccharides (GOS)) [44]. β-Glucans are the principal soluble fibers found in abundance in oat grain (free of FODMAPs) and barley or wheat. Numerous studies have shown that β-glucans can significantly stimulate several types of immune responses [44]. Recently, Zyla et al. showed that β-glucans reduce the gene expression of proinflammatory cytokines in the colon in rat models with trinitrobenzene sulfonic acid (TNBS)-induced CD [50].

## 4. Conclusions

To the best of our knowledge, there is no study that has examined the effects of the MedDiet—full of fruits, vegetables, and legumes, and therefore rich in FODMAPs—on IBS patients, thus far. IBS is characterized by low-grade inflammation restricted to the bowels, and the LFD alleviates IBS symptoms, but it is ineffective in inflammatory response management. On the contrary, the MedDiet has anti-inflammatory and immunomodulating properties and increases the abundance of bacterial species in the intestine that are beneficial for the host. Thus, the MedDiet is not only considered a prudent dietary pattern, but also a scientifically proven method that is able to provide benefits for the management of several diseases and an overall improvement in health, while the LFD might be considered a possibility to identify, if present, some intolerances. Recently, the beneficial effect of supplementing the LFD with either dietary fiber or probiotics has been reported, limiting constipation and dysbiosis, respectively; thus, similar effects might be detected with adequate consumption of allowed fruits and vegetables rich in prebiotics, abundant in the MedDiet (Table 2).

The novelty of this review is the concept of a new dietary pattern with the synergistic action of the LFD and MedDiet as a dietary treatment for IBS. We speculate that enriching the LFD with the MedDiet’s components with anti-inflammatory and prebiotic actions may potently optimize the effects of the LFD and eliminate its drawbacks.

**Table 2 microorganisms-10-00751-t002:** Common foods in the MedDiet, free of FODMAPs according to the Monash University application.

Food	Components	Action
Oat	β-Glucans (fibers)	prebiotic action [51]
Olives/olive oil	Hydroxy-tyrosol (phenolic compound)	anti-inflammatory action [44]
Walnuts	ω-3 PUFAs	prebiotic action [52]
Fish	ω-3 PUFAs	anti-inflammatory action [53]
Wine	Resveratrol (phenolic compound)	anti-inflammatory action [54]
Orange	Quercetin (phenolic compound)	prebiotic action [55] anti-inflammatory action [56]
Mandarin (Imperial, clementine)	Quercetin (phenolic compound)	anti-inflammatory action [57]
Tomatoes	Quercetin (phenolic compound)	prebiotic action [58]
Oregano, rosemary, thyme, and cumin	Phenolic compounds	anti-inflammatory action [59]

Prebiotics promote the growth of beneficial bacteria and the development of intestinal epithelial cells while reshaping the gut microbiota. Intake of VOO, fish, and specific fruits and vegetables and moderate wine consumption according to the MedDiet pyramid, as well as limited intake of red meat, proceed foods (i.e., bacon, sausages), and refined oils, in all phases of the LFD could be crucial to reduce the inflammatory status.

The limitation of this new concept is the lack of evidence from clinical trials to support it. However, there is evidence that the MedDiet is beneficial for the symptom management of patients with quiescent (absence of laboratory and endoscopic findings of inflammation) IBD—the so-called IBS-like syndrome [60]. Moreover, the concept of a new dietary pattern with the synergistic action of the LFD and MedDiet as a dietary intervention for IBS is currently under evaluation in the ClinicalTrials.gov Identifier: NCT03997708 study, where we compare the efficacy of a combination of the MedDiet and the LFD and the nutritional recommendations of NICE in managing IBS. Additionally, the collected fecal samples will be used to assess the effect of each intervention on the gut microbiota, which might shed light on the underlying mechanisms of the additive or synergistic effect of the two dietary interventions.

## Figures and Tables

**Figure 1 microorganisms-10-00751-f001:**
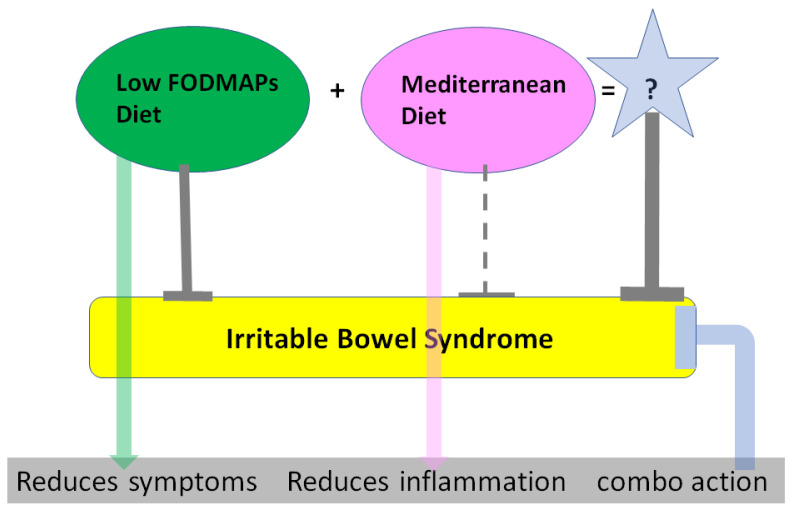
Schematic presentation of the hypothesis of combining the Mediterranean diet with the low-FODMAP diet for the management of irritable bowel syndrome.

**Figure 2 microorganisms-10-00751-f002:**
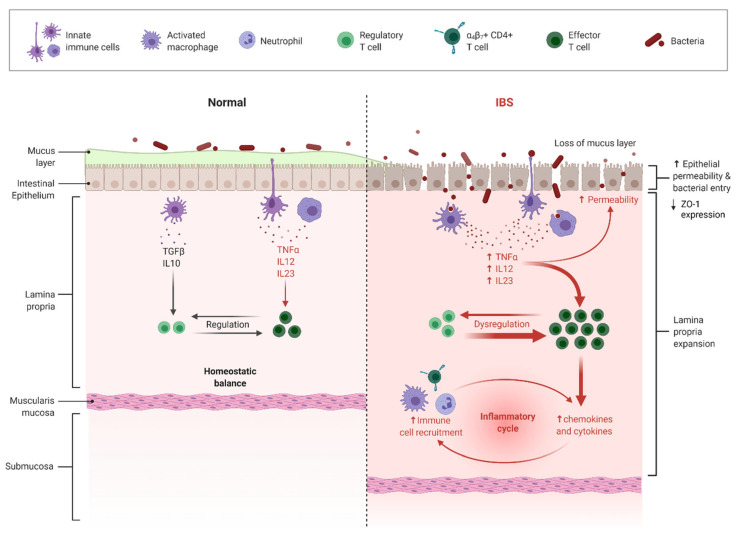
Intestinal epithelium in healthy and IBS patients. Figure 2 was created using BioRender, Toronto, Canada, Subscription agreement number: TU237PZA2O for *Microorganisms*.

**Table 1 microorganisms-10-00751-t001:** Effect of the LFD on gut microbiota [7,8,29,30].

Effect of the LFD on Gut Bacteria
*Bifidobacterium (Actinobacteria)*	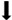
*Bacteroides*	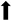
*Lactobacillus*	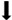
*Clostridium* cluster XIVa	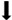
*Clostridium* cluster IV	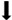
*Faecalibacterium prausnitzii*	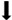
*Mycoplasma hominis*	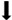
*Mycoplasma hominis*	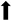

## Data Availability

Not applicable.

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
