# Peer review of "A Combination of Mediterranean and Low-FODMAP Diets for Managing IBS Symptoms? Ask Your Gut!"

_microorganisms, 2022, doi:10.3390/microorganisms10040751_

Round 1

Reviewer 1 Report

Relevant study with broad appeal, would ask for some additional thoughts about next steps to look into mechanism for possible additive or synergistic effects of overlap between low FODMAP and Mediterranean diets. 

Author Response

Reviewer 1

Relevant study with broad appeal, would ask for some additional thoughts about next steps to look into mechanism for possible additive or synergistic effects of overlap between low FODMAP and Mediterranean diets. 

Response: Indeed, the concept of a new dietary pattern with synergistic action of LFD and MedDiet as a dietary treatment for IBS is currently under evaluation in ClinicalTrials.gov Identifier: NCT03997708 study, where we evaluate the efficacy of a combination of MedDied+LFD and the nutritional recommendations of NICE in managing IBS. Additionally, the collected fecal samples will be used to assess the effect of each intervention on gut microbiota, which might seed light in the underlying mechanisms of the additive or synergistic effect of the two dietary interventions  

We included this statement at the end of the conclusion section

Reviewer 2 Report

The review is interesting. Given that no studies have directly examined the effect of Mediterranean diet on managing IBS symptoms, the title is misleading and needs to be revised.

Furthermore, the authors can speculate potential beneficial effects of a Mediterranean diet, but given that there are no studies to demonstrate this effect, the conclusions are purely speculative. This needs to be made clear and discussed as a caveat.

Author Response

Reviewer 2

The review is interesting. Given that no studies have directly examined the effect of Mediterranean diet on managing IBS symptoms, the title is misleading and needs to be revised.

Response: We revised the title as:  A combination of Mediterranean and Low FODMAP diet for managing IBS symptoms? Ask your gut. 

Furthermore, the authors can speculate potential beneficial effects of a Mediterranean diet, but given that there are no studies to demonstrate this effect, the conclusions are purely speculative. This needs to be made clear and discussed as a caveat.

Response:  This caveat is now highlighted in the conclusion, and it is discussed as limitation at the last paragraph of the conclusions 

Reviewer 3 Report

This short review is mostly dedicated to the Mediterranean Diet (MD).

Reading the text, all the data reported underline the importance of MD in terms of health and well-being.

In summary, the conclusions are that, in case of IBS diagnosis, the adoption of the FODMAP diet must be integrated with MD to effectively counteract the symptoms of IBS, in all its variants. This is what your gut is asking for!

I share the point of view of the authors. However, I suggest the authors to improve some parts that are quite confusing (see below) and better emphasize that IBS is not a functional disease. This is also in agreement with the efficacy of the MD which seems to act mainly for its anti-inflammatory activity, as the authors report. Thus suggesting that in IBS there is an inflammatory state. I also suggest, in conclusion, to clearly highlight the general importance of MD to guarantee a good state of health while the FADMAP is only a possibility to identify, if present, some intolerances.

  1. Row 46-50 It is not clearly explained the relation between IBS and IBD in term of symptoms what are the IBS-like IBD and the IBS-like symptoms and histopathological signs. If the Intestinal barrier is altered how to distinguish between IBS or IBD?
  2. Row 60. The references quoted are limited. Foe example the authors could quoted a review published in J Clin Med, 2018 by Vannucchi and Evangelista.
  3. Rows 109-121: Please make order and clarity in this description of the data… The paragraph is confusing.

Author Response

Reviewer 3

This short review is mostly dedicated to the Mediterranean Diet (MD). Reading the text, all the data reported underline the importance of MD in terms of health and well-being. In summary, the conclusions are that, in case of IBS diagnosis, the adoption of the FODMAP diet must be integrated with MD to effectively counteract the symptoms of IBS, in all its variants. This is what your gut is asking for!

I share the point of view of the authors. However, I suggest the authors to improve some parts that are quite confusing (see below) and better emphasize that IBS is not a functional disease. This is also in agreement with the efficacy of the MD which seems to act mainly for its anti-inflammatory activity, as the authors report. Thus, suggesting that in IBS there is an inflammatory state. I also suggest, in conclusion, to clearly highlight the general importance of MD to guarantee a good state of health while the FADMAP is only a possibility to identify, if present, some intolerances.

Response: According to your request we emphasize in the revision that IBS is NOT a functional disorder by modifying the first, second and fourth paragraph of the introduction. Moreover, we added the different roles of the two diets in the first paragraph of the conclusion, as requested.

  1. Row 46-50 It is not clearly explained the relation between IBS and IBD in term of symptoms what are the IBS-like IBD and the IBS-like symptoms and histopathological signs. If the Intestinal barrier is altered how to distinguish between IBS or IBD?

Response: We deleted these sentences because they created confusion. We also added indirect from IBS-like management evidence that MedDiet might be efficacious for IBS in the conclusions section, page 9, lines 289-291 (ref 60)

  1. Row 60. The references quoted are limited. For example the authors could quoted a review published in J Clin Med, 2018 by Vannucchi and Evangelista.

Response: Thank you, reference added (Ref number: 13)

  1. Rows 109-121: Please make order and clarity in this description of the data… The paragraph is confusing.

Response: Paragraph edited to be more comprehensive, as requested

Round 2

Reviewer 3 Report

I appreciated the efforts the authors made in considering the reviewer' comments.